# Observation of Abrupt Changes in the Sea Surface Layer of the Adriatic Sea

Frano Matić [1,2,*], Tomislav Džoić [1], Hrvoje Kalinić [3], Leon Ćatipović [3], David Udovičić [1], Tea Juretić [1], Lucija Rakuljić [2], Daria Sršen [2] and Vjekoslav Tičina [1]

1    Institute of Oceanography and Fisheries, 21000 Split, Croatia; dzoic@izor.hr (T.D.); udovicic@izor.hr (D.U.); juretic@izor.hr (T.J.); ticina@izor.hr (V.T.)
2    University Department of Marine Studies, University of Split, 21000 Split, Croatia; lucija.rakuljic7@hotmail.com (L.R.); dariasrsen31@gmail.com (D.S.)
3    Faculty of Science, University of Split, 21000 Split, Croatia; hrvoje.kalinic@pmfst.hr (H.K.); leoncatip@pmfst.hr (L.Ć.)
*    Correspondence: fmatic@izor.hr

**Abstract:** We observed interannual changes in the temperature and salinity of the surface layer of the Adriatic Sea when measured during the period 2005–2020. We observed non-stationarity and a positive linear trend in the series of mixed layer depth, heat storage, and potential energy anomalies. This non-stationarity was related to the climate regime that prevailed between 2011 and 2017. We observed significant changes in the interannual variability of salinity above and below the mixed layer depth and a positive difference in the surface barrier layer. In an effort to reconstruct the cause of this phenomenon, a multi-stage investigation was conducted. The first suspected culprit was the change in wind regime over the Mediterranean and Northeast Atlantic regions in September. Using the growing neural gas algorithm, September wind fields over the past 40 years were classified into nine distinct patterns. Further analysis of the CTD data indicated an increase in heat storage, a physical property of the Adriatic Sea known to be strongly influenced by the inflow of warm water masses controlled by the bimodal oscillating system (BiOS). The observed increase in salinity confirmed the assumption that BiOS activity affects heat storage. Unexpectedly, this analysis showed that an inverse vertical salinity profile was present during the summer months of 2015, 2017, and 2020, which can only be explained by salinity changes being a dominant factor. In addition, the aforementioned wind regime caused an increase in energy loss through latent energy dissipation, contributing to an even larger increase in salinity. While changes in the depth of the mixed layer in the Adriatic are usually due to temperature changes, this phenomenon was primarily caused by abrupt changes in salinity due to a combination of BiOS and local factors. This is the first record of such an event.

**Keywords:** mixed layer depth; climate regimes; heat storage; sea surface layer; heat fluxes; neural gas; wind patterns

## 1. Introduction

Ocean–atmosphere interaction is a major cause of climate variability and is affected by climate change through global warming; energy exchange through air–sea processes; and changes in planetary, regional, and local meteorological processes [1]. Anthropogenic greenhouse gas emissions have resulted in an energy imbalance of about 0.4–1 W/m$^2$ on Earth, most of which accumulates as heat in the ocean [2]. During the summer months, the ocean water forms a vertical, stable stratification with warmer and lighter water near the surface, which significantly affects the vertical mixing of heat in the water column. Upper ocean stratification has recently increased by 5.3% globally (at a rate of 0.90% per decade), primarily due to temperature changes, with salinity playing an important role locally [3].

Over the last decade, the effects of climate change on biotic and abiotic parameters in the Adriatic Sea have been noted. The sea surface temperature increased by 1 °C between 1979 and 2015, with significant multi-decadal variations [4]. From 2008, the temperature increase accelerated and showed a linear trend of 0.013 °C/month, affecting the microbial food web [5]. Temperature changes affect vertical mixing and stratification, as well as altering the nitrogen cycle and food web in the ocean [6]. The superposition of multi-decadal thermohaline fluctuations, low river discharge, dry weather conditions, and excessive evaporation relative to precipitation caused sub-surface salinity to reach a historical maximum in 2017 [7].

One of the factors affecting the physical and chemical water properties of the Adriatic Sea, along with river inflows and meteorological conditions, is the bimodal oscillating system (BiOS) [8–12]. The BiOS mechanism refers to the upper layer cyclonic/anticyclonic circulation in the Ionian Sea, which influences exchange with the Adriatic Sea (and vice versa) on a quasi-decadal scale. Advection of the less saline Atlantic water (AW) occurs during the anticyclone phase, while salty Levantine intermediate water (LIW) is introduced from the Ionian Sea into the Adriatic Sea during the cyclonic phase [13]. From satellite altimetry measurements in the northern Ionian Sea, occurrences of BIOS phases can be identified in the last 23 years as cyclonic (1998–2005, 2011–2017) and anticyclonic (2006–2010, 2018–2019) [8,14].

The synoptic activity over the Adriatic Sea, as well as in the parent Mediterranean basin, has a well-defined annual cycle, with a storm season that encompasses the more intense period from November to March. Three main seasons of winter, summer, and spring can be identified, as the features of autumn months can be characterised as either late summer or early winter [15]. The Gulf of Genoa is one of the major cyclogenetic areas from which cyclones travel southeastward, affecting the Adriatic Sea [16]. Although winter is the period in which cyclones mostly occur in the gulf of Genoa, summer and autumn cannot be excluded. During the summer months (June, July, and August) most cyclones that appear in the Adriatic Basin (and their tracks) are classified as Genoa cyclones, with a smaller number of Adriatic cyclones. Autumn shows a different picture, with equal numbers of Genoa, Adriatic, and non-Genoa and -Adriatic cyclones [17]. Due to the summer and autumn pressure gradient between the Azores High and the Persian Low over the Mediterranean, there are large-scale flow systems called the Etesian winds. Over the Adriatic, they are superimposed on the local sea/land breeze circulation [18,19], which results in a local summer wind called the Maestral. Other characteristic winds that can have an impact during the transition from summer to autumn are the Adriatic Sirocco and Bora. The persistent south-to-southeasterly wind of Sirocco, confined to the Adriatic Basin by the Apennines and Dinaric Alps, is usually generated by Genoa cyclones or anticyclones over the Mediterranean [20]. The Bora is northeasterly downslope wind along the east Adriatic coast, generated by anticyclones to the north or northeast or cyclones southwest of the Adriatic region [21]. Taking into account the analysis of regional climate model simulations, it has been shown that, in summer, the southern Adriatic is under the prevailing regime of Etesian, with great persistence (steadiness > 60%), while the northern Adriatic is under the regime of sea/land breezes and slope winds (steadiness < 30%), which alternate direction in 24, 12, 8 and 76 h periods [22].

The mixed layer depth, as the boundary between the well-mixed surface layer and the intermediate waters, is modified by oceanic processes (advection and internal waves) and heat and momentum exchange with the atmosphere, which is controlled by local and global meteorological processes on a time scale of seconds to several years. Because MLD is a key parameter for phytoplankton community distribution and light penetration its changes are important for understanding the effects of climate change on the ocean ecosystem.

Measurements of temperature, salinity, and depth of the mixed layer in the water column are an important activity in hydroacoustic surveys to determine the characteristics of the acoustic field in which acoustic targets are observed. The absorption of sound energy and the vertical profile of sound velocity depend on the thermohaline properties of the

water column [23]. Therefore, vertical measurements with a calibrated CTD sonde are used to update the parameters of scientific echosounders during acoustic surveys and to collect accurate acoustic data on acoustic targets, which are recorded in echograms. In addition, environmental parameters affect the spatial distribution of small pelagic fishes and, consistent with the survey protocol, a network of CTD stations will also be sampled during hydroacoustic surveys to describe important oceanographic features of the area under study [24].

The remainder of this manuscript is organised as follows: Section 2 presents the data from local to global scale. We detial the methods used, starting with the growing neural gas algorithm. The results consider the thermodynamic properties of vertical profiles of temperature and salinity, in order to determine their temporal changes and trends. Subsequently, the neural gas method is applied to wind data, mean sea level pressure, and NAO over the Mediterranean Sea.

## 2. Materials and Methods

### 2.1. Data

Temperature and salinity data were collected with oceanographic (CTD) probes during eighteen acoustic surveys, performed mainly in September between 2005 and 2020, with a low sampling density (about 43 stations/survey) before 2012 and a high sampling density (about 90 stations/survey) after 2012 (Figure 1, Table 1). A total of 1136 different CTD measurements were collected with several Seabird 25 and one Idronaut 316 CTD probes, all with an accuracy greater than $\pm 0.002\,°C$ for temperature and $\pm 0.002$ for salinity. To maintain sampling homogeneity and optimal CTD data performance, the CTD probes were calibrated and intercalibrated with each other at regular intervals before or after the survey. The CTD probes collected data at a sampling frequency of 8 Hz, resulting in a final vertical resolution of 0.5 m. To maintain mutual homogeneity, all profiles and metrics were processed according to the procedure recommended by the instrument manufacturer. These included: median filter with a window size of 10 s (80 bins) for spike detection, averaging of measurements at a vertical resolution of one meter, and visual quality control.

**Table 1.** Basic oceanographic information of acoustic surveys conducted during 2005–2020, including number of CTD casts together with research area (marked as minimum–maximum latitude and longitude).

| | Cruise | | CTD | Latitude [°N] | | Longitude [°E] | |
|---|---|---|---|---|---|---|---|
| Year | Start | End | Cast | Min | Max | Min | Max |
| 2005 | 20 August | 4 September | 43 | 42.306500 | 45.325667 | 13.252333 | 18.414000 |
| 2006 | 2 September | 18 September | 43 | 42.352000 | 45.324500 | 13.147667 | 18.460167 |
| 2007 | 31 August | 17 September | 35 | 42.349833 | 45.380667 | 13.327500 | 18.464167 |
| 2008 | 1 September | 18 September | 43 | 42.585000 | 45.318667 | 13.252000 | 18.119667 |
| 2009 | 21 September | 14 November | 43 | 42.342167 | 45.373333 | 13.235333 | 18.483500 |
| 2010 | 2 September | 18 September | 48 | 42.325983 | 45.336367 | 13.130117 | 18.451183 |
| 2011 | 5 September | 18 September | 49 | 43.101283 | 45.430483 | 13.078833 | 16.390667 |
| 2012 | 3 September | 18 September | 48 | 43.403000 | 45.473667 | 13.019833 | 16.390667 |
| 2013 | 3 September | 29 September | 88 | 42.299500 | 45.507333 | 13.066667 | 18.527000 |
| 2014 | 4 September | 30 September | 90 | 42.315900 | 45.510083 | 13.061350 | 18.403533 |
| 2015 | 4 September | 5 October | 88 | 42.305333 | 45.494167 | 13.061500 | 18.427667 |
| 2016 | 24 August | 21 September | 93 | 42.340380 | 45.467690 | 13.057620 | 18.518370 |
| 2017 | 30 August | 30 September | 89 | 42.299917 | 45.505083 | 13.059733 | 18.523283 |
| 2018 | 29 August | 23 September | 90 | 42.304167 | 45.504833 | 13.062333 | 18.523833 |
| 2019 | 28 August | 28 September | 91 | 42.300042 | 45.499512 | 13.048550 | 18.524533 |
| 2020 | 28 August | 5 October | 88 | 42.300905 | 45.498322 | 13.033302 | 18.524583 |

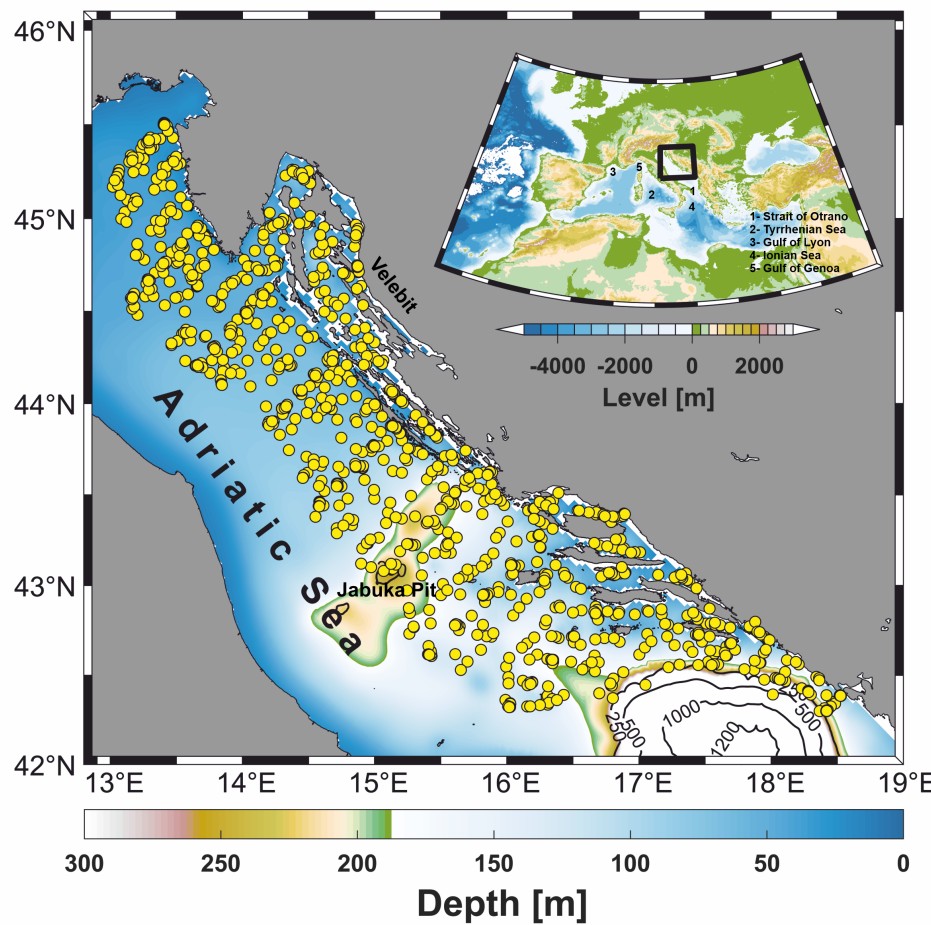

**Figure 1.** Adriatic Sea with investigated oceanographic sampling site areas (marked with circles).

The use of climate re-analysis products is an indispensable tool in the study of climate characteristics and changes, as re-analysis combines atmospheric models with all available observations to obtain the most accurate and comprehensive numerical estimate of past climate data. ERA5 is produced using 4D-Var data assimilation and provides atmospheric, land surface, and ocean wave variables at a horizontal resolution of 31 km with hourly output [25,26]. The used ERA5 variables included hourly 10 m wind vector, mean sea level pressure (MSLP), surface latent heat flux, surface net solar radiation, surface net thermal radiation, and surface sensible heat flux daily data at 12:00 for the period 1981–2020, for the months of June, July, August, and September, at a spatial resolution of 0.25° latitude and 0.25° longitude in the 20 W–40 E and 25 N–55 N coverage areas, covering the period from 1981 to 2020 ([27]; https://cds.climate.copernicus.eu, accessed on 15 March 2021).

### 2.2. Methods

#### 2.2.1. Synoptic Field Analysis Methods

To extract characteristic wind time patterns, the growing neural gas network (GNG) method was used. GNG is a clustering method, somewhat similar to the self-organising maps algorithm [28], which has been used in the ocean sciences field on numerous occasions [6,29,30]. Both are trained through unsupervised learning. While self-organising maps (SOM) require prior knowledge of the topological structure of the data manifold to be efficient and useful, GNG is more flexible in this respect (i.e., it does not require any prior knowledge of the network topology to learns the similarity relations between input signals). As a result, GNG reduces the dimensionality of the data space to an arbitrary number of neurons (best matching units; BMUs) by assigning them to the relevant parts of the data space while minimising the representation error. Unlike the strong interneural

connections in SOM, GNG places no emphasis on the strength of the connections between BMUs, allowing them to propagate "freely" through the data space, mimicking the behaviour of gas molecules [31]. The wind and MSLP input data had to be reshaped and organised in a particular way, as the algorithm is sensitive to the input shape. Indeed, a simple transposition of the input data leads to very different physical interpretations of the output. Therefore, the wind data had to be arranged such that the rows represent time instances, while the two columns represent the collective spatial ranges of wind components $u$ and $v$, respectively. To achieve this data arrangement, the original data had to be merged on an hourly scale within the abovementioned geographical boundaries for the $u$ and $v$ components. The resulting 2D arrays of data at a given time $t$ were flattened in the order of the main series before being stacked and assembled. At the end of this single step, the data had the form of (1, 63,162), or (one hour, (latitude points × longitude points) × 2). This procedure was then repeated for every hour of every day of every September from 1979 to 2019, resulting in an array with dimensions (29,520, 63,162). The data from MSLP were transformed in the same way, resulting in an array of the form (29,520, 31,581), as there was only a single pressure component.

The formatted data were fed into the GNG algorithm of the Python library NeuPy. As mentioned earlier, the parameters of the algorithm were fixed: Step = 0.1, neighbour step = 0.001, maximum edge age = 50, number of iterations before adding a neuron = 100, after-split error decay rate = 0.5, error decay rate = 0.995, and minimum update distance = 0.2. The number of desired nodes was set to 9, corresponding to the number of resulting BMUs. The training duration was set to 3000 epochs. By comparing the BMUs and the formatted data mentioned above, we extracted the temporal order of the dominantly active BMUs using the least vector norm. Finally, with the corresponding temporal order of the BMUs, we were able to generate an average pressure field for each BMU using the MSLP data that we had previously formatted.

### 2.2.2. Sea Data Analysis Methods

The mixing layer (ML) can be defined as a surface layer with constant potential density, or a surface layer with constant temperature. The depth at which the density begins to increase is the mixed layer depth (MLD). The MLD was defined as the depth at which the potential density increases corresponding to a temperature decrease of 0.5 °C, while salinity and pressure are held constant. The depth at which the temperature begins to decrease is referred to as the isothermal layer depth (ILD). The ILD was determined to be the depth at which the temperature was 0.5 °C lower than that prevailing between a depth of 2 and 6 m on average. The required increase in potential density was approximately $0.15 \, \text{kg} \cdot \text{m}^{-3}$. By using these definitions, we were able to avoid the effects of daily cycles of MLD/ILD for time-averaged fluxes through the base ML. Using the ILD criterion of 0.2 °C (which corresponds to an MLD criterion of $\approx 0.06 \, \text{kg} \cdot \text{m}^{-3}$), diurnal cycles occurred. No significant difference between the MLD and ILD was observed in the high-resolution glider data at any site. Therefore, the simpler temperature criterion was chosen for our definition of MLD [1]. The mixed layer depth (MLD) was calculated using [32,33]. Reducing the time-series data to monthly median values, the procedure used in constructing the climatology (Section 2.2) showed that the 0.2 °C MLD criterion remained a good estimator of the mixed layer depth envelope. The choice of the MLD temperature criterion, especially with respect to the choice of [34] and the classical 0.5 °C threshold from [35], was further verified by comparison with several anchored time-series. Their high temporal resolution at a fixed point contrasted with the large number of profiles having widely temporally distributed climatology. These comparisons indicate that the 0.2 °C threshold criterion calculated from the 10-m temperature is quite successful in estimating the MLD, and captures the first spring re-stratification [36] particularly well.

The heat storage (HS) in the water column between the surface and the depth of the mixed layer is calculated using

$$\text{HS} = c_p \rho \int_0^{h_{MLD}} T(z)dz, \tag{1}$$

where $c_p$ is the specific heat capacity of the seawater, $\rho$ is the density of water, and $T(z)$ is the vertical profile of the self-measured CTD measurement [37].

Simpson and Bowers [38] have defined the potential energy anomaly (PEA) as

$$\text{PEA} = \frac{1}{H} \int_0^H (\rho(z) - \rho)gzdz, \tag{2}$$

where $z$ is the vertical coordinate, $H$ is the bottom depth, $\rho(z)$ is the density evaluated at $z$, and $\rho$ is the mean density of the vertical profile. The potential energy anomaly becomes positive for a stable stratified water column and negative for an unstable stratified water column. Physically, this indicates the amount of energy per volume required to mix the entire water column and achieve complete vertical mixing.

The total air–sea energy exchange ($Q$) was calculated as $Q = Q_{Sh} + Q_{Lo} + Q_{La} + Q_{Se}$, where $Q_{Sh}$ represents the shortwave solar radiation incident on Earth, $Q_{Lo}$ the backward longwave radiation, followed by $Q_{La}$ and $Q_{Se}$, the latent and sensible turbulent energy. We used the European Centre for Medium-Range Weather Forecasts (ECMWF) convention for vertical fluxes, where fluxes are positive downward.

The different climate regimes in the time-series were determined using the sequential *t*-test analysis of regime shifts (STARS; [39–42]) algorithm. Before applying the STARS algorithm, a pre-whitening tool was used to filter out autocorrelation from the time-series. The detection of regime shifts (RS) in the mean was performed with a target significance level $p = 0.01$ and cut-off length $l = 300$.

## 3. Results

The water masses below the mixed layer were determined using a temperature–salinity diagram (T–S diagram; see Figure 2). As can be seen from the diagram, dense water formed during the past winters along the eastern Adriatic coast [43] and the northern and southern Adriatic Sea, spreading over the seafloor and following the ocean bathymetry to the Jabuka Pit and the Strait of Otranto [44,45]. The origin of the water, however, differed. The coldest and less saline water masses came from the northeastern Adriatic coast, an area influenced by the Velebit Mountains, with its numerous submarine springs, and the strong katabatic Bora wind. These water masses formed during strong Bora episodes, followed by strong evaporation in January and February. One of these episodes during our studied period included the strongest cooling event in February 2012 [46]. The strongest consequence of these winter Bora wind episodes in the shallow northern Adriatic was the formation of dense water masses that later reached the bottom of the Jabuka Pit and the South Adriatic Pit. The warmest and most saline water mass on the diagram was introduced from the Ionian Sea. An important feature of the diagram is that the salinity in the mixed layer was the highest in the history of measurements in the Adriatic Sea, accompanied by strong interannual variations [7].

The thermohaline properties of the HS, MLD, and PEA time-series exhibited a strong temporal and meridional signal, with substantial variations in multi-year variability (Figure 3). Due to the characteristics of the measurements, the variations in the meridional signal had a strong temporal component in addition to the station depth, which is related to the fact that the measurements were made over a period of twenty days or less. It was found that HS accumulates 2.06 MJ on average in the mixed layer with an average depth of 13.82 m, while the PEA was calculated as 20.41 J/M$^3$ for the entire water column.

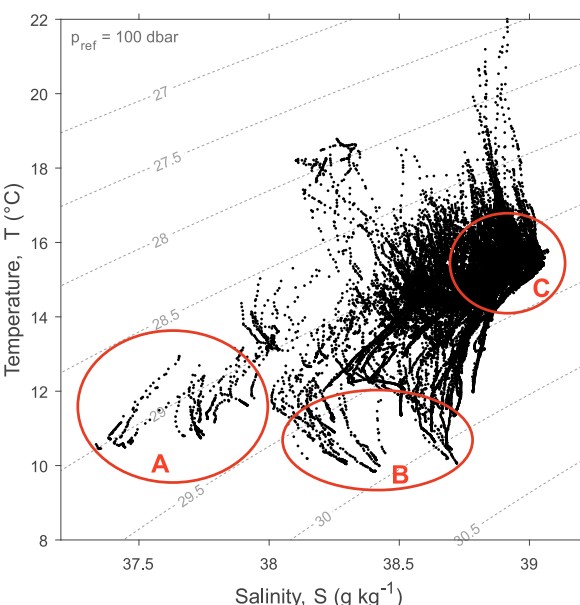

**Figure 2.** Temperature–salinity diagram with isopycnals (dotted grey line) plotted for measurements deeper than 100 m (black dots). Data cover the period from 2005 to 2020. The red ellipse shows (**A**) water masses in the Velebit Channel, (**B**) the Jabuka Pit and (**C**) Levantine intermediate water.

No significant trend was observed for the HS, MLD, and PEA, mainly as the climate regime occurred in the middle of the period. The climate regime was detected by the STARS algorithm, and was responsible for classification of the series as non-stationary. The variable for the break in stationarity between the series started between 2011 and 2013, and an increase of 0.62 MJ in the HS was observed between 2011 and 2017. This increase was not tracked by the MLD and PEA, which entered a new climate regime two years after the HS, increasing by 5.39 m and 2.594 $J/m^3$, respectively. The PEA returned to the original climate regime in 2017, but the HS and MLD entered a new regime in 2018 with a higher mean.

Changes were also detected in the surface barrier layer (SBL), as indicated by the difference between the MLD and ILD (Figure 4c). The SBL determines the influence of salinity on the stratification of the boundary layer. In the Adriatic and Mediterranean Sea, the influence of salinity on SBL is less important than that of temperature, resulting in values around zero most of the time. SBL values below zero were found at the stations near the coast or under the influence of river discharge. The main result was slightly positive SBL values, showing salinity vertical distribution as the main factor for MLD. Anomalous SBL was recorded during 2013–2017 and 2020, with an increasing trend that reached its maximum in 2017. A significant linear trend (Figure 4a), with an increase of 2.1 decade$^{-1}$, was observed for salinity below the MLD. In the Adriatic Sea, salinity inversion is characteristic of the surface layer (Figure 4b). In September 2015, 2017 and 2020, the inversion was replaced by an opposite trend of stratification, with higher salinity in the boundary layer than below the boundary layer.

The MLD changed over the long run, but the climate regime persisted for more than six years, likely because planetary atmospheric and oceanographic processes contribute to this multi-year mechanism. A study of wind patterns over the Mediterranean Sea and energy exchange over the Adriatic Sea revealed the causes of these thermocline depth variations.

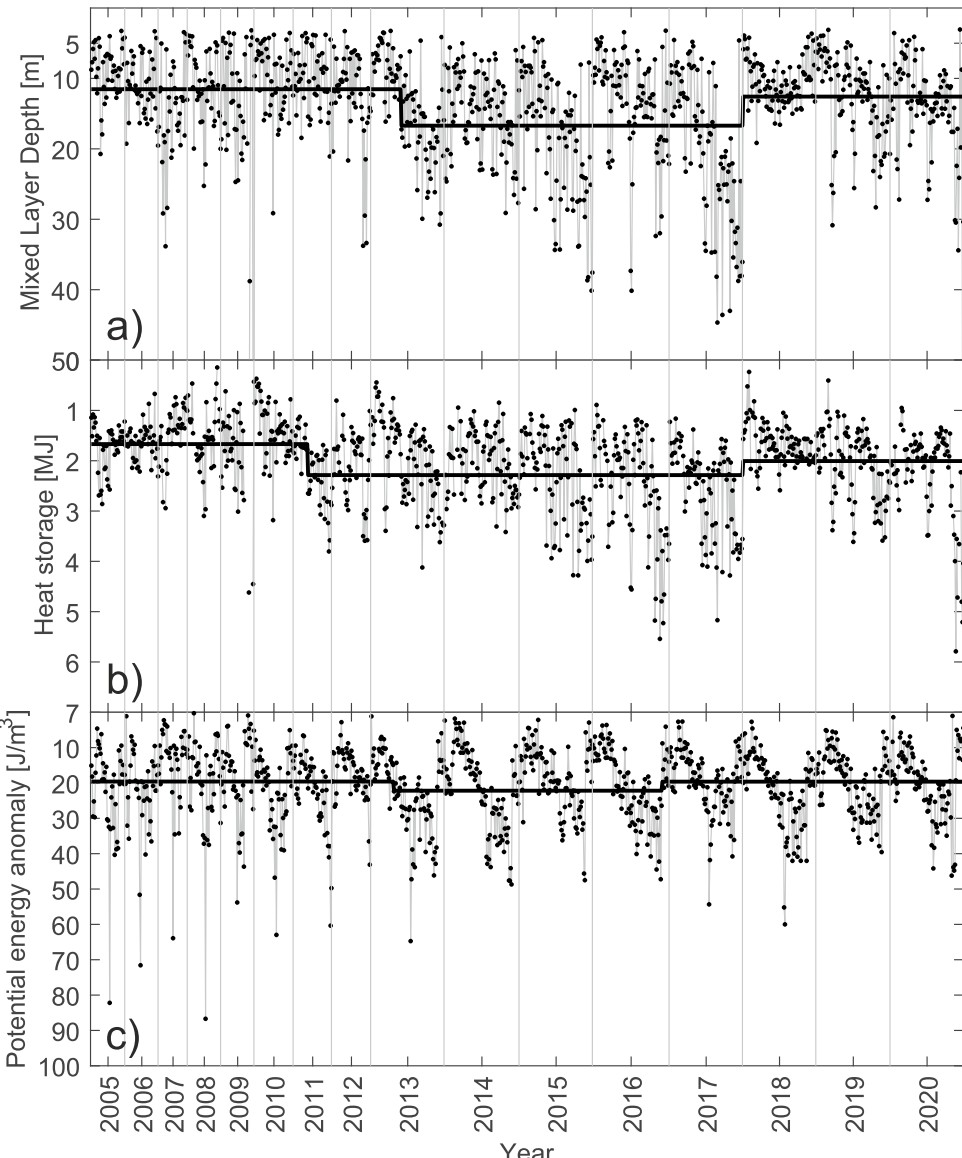

**Figure 3.** Mixed layer depth (**a**); heat storage (**b**); and potential energy anomaly (**c**) calculated from the vertical CTD profiles in September between 2005 and 2020. The bold line shows the climate regime of the time-series. Note that the *x*-axis is not uniformly distributed.

By classifying wind speeds over the Mediterranean Sea and western Atlantic Ocean using GNG, we could identify the average and extreme wind patterns responsible for variations in the oceanographic features of the Adriatic Sea (Figure 5). With GNG, the wind field was classified into nine clusters and the average MSLP was calculated for each cluster. Based on the wind speed and MSLP patterns, the meteorological conditions are described as follows. Low pressure continued to prevail over Asia Minor in September, with an exchange of high pressure (BMU 2, 3 and 4) and low pressure (BMU 1, 5 and 9) over central and western Europe. A strong north–south MSLP gradient over the Levantine Basin results in NW winds over the basin (BMU 2, 3, 4, 6 and 8). The Adriatic is not part of the main Etesian wind pattern that occurs with varying wind speeds, very slow NW winds (BMU 2), and the strongest NE–NW wind pattern in BMU 4 and 6. On the other hand, BMU 7 and 9 are associated with cyclonic activity in the Tyrrhenian, Ionian, and Adriatic Seas, causing strong winds over the Gulf of Lyon and complex winds in the Adriatic Sea. In BMU 7, the minimum of MSLP is located in the southern Adriatic and northern Ionian Seas with cyclonic wind patterns (NE in the northern Adriatic and SE in the southern Adriatic),

while in BMU 9, the minimum of MSLP is located over the central Adriatic, resulting in NW wind patterns. BMUs 1 and 5 show a south–north gradient over the Adriatic, leading to strong winds from SE (Sirocco wind).

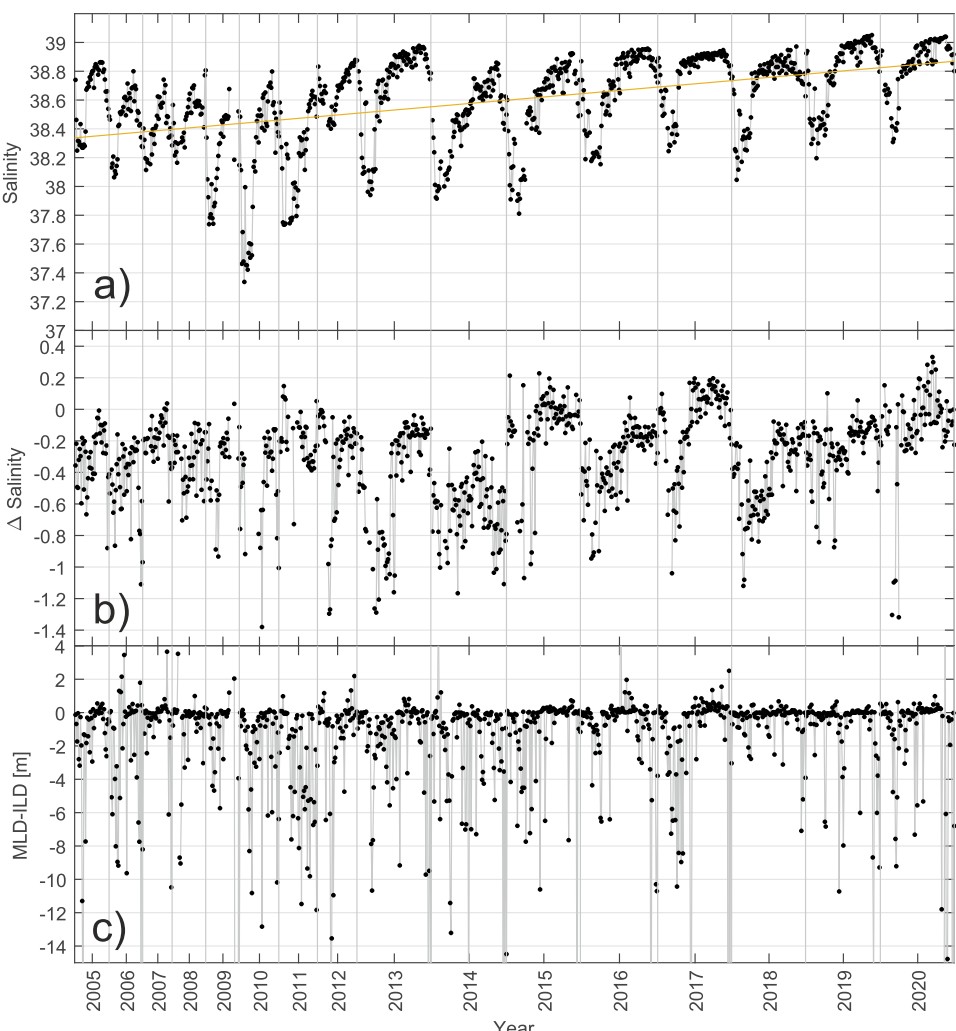

**Figure 4.** (**a**) Mean salinity below mixed layer depth (MLD); (**b**) the difference between mean salinity above and below mixed layer depth; and (**c**) the difference between mixed layer depth and isothermal layer depth (ILD), calculated from vertical CTD profiles in September between 2005 and 2020. The yellow line shows a significant ($p < 0.01$; [47,48]) linear trend. Note that the *x*-axis is not uniformly distributed.

Typical patterns (BMU 1–9) were extracted along with more and less extreme events that had a uniformly distributed occurrence frequency of 136 days. The quality and distribution of the modelled BMUs were visualised using principal component analysis (PCA) of the September wind data from 1979 to 2019 with [49]. BMUs and data were projected onto a PC1–PC2 grid, with PC1 and PC2 accounting for 13% and 10% of the total variance, respectively (Appendix A Figure A1). Note the circular distribution of data points around the origin. The most extreme patterns were BMU 1, 6 and 8. Furthermore, BMU 1 and 3 were associated with a positive North Atlantic Oscillation (NAO) phase, while other teleconnection indices were not.

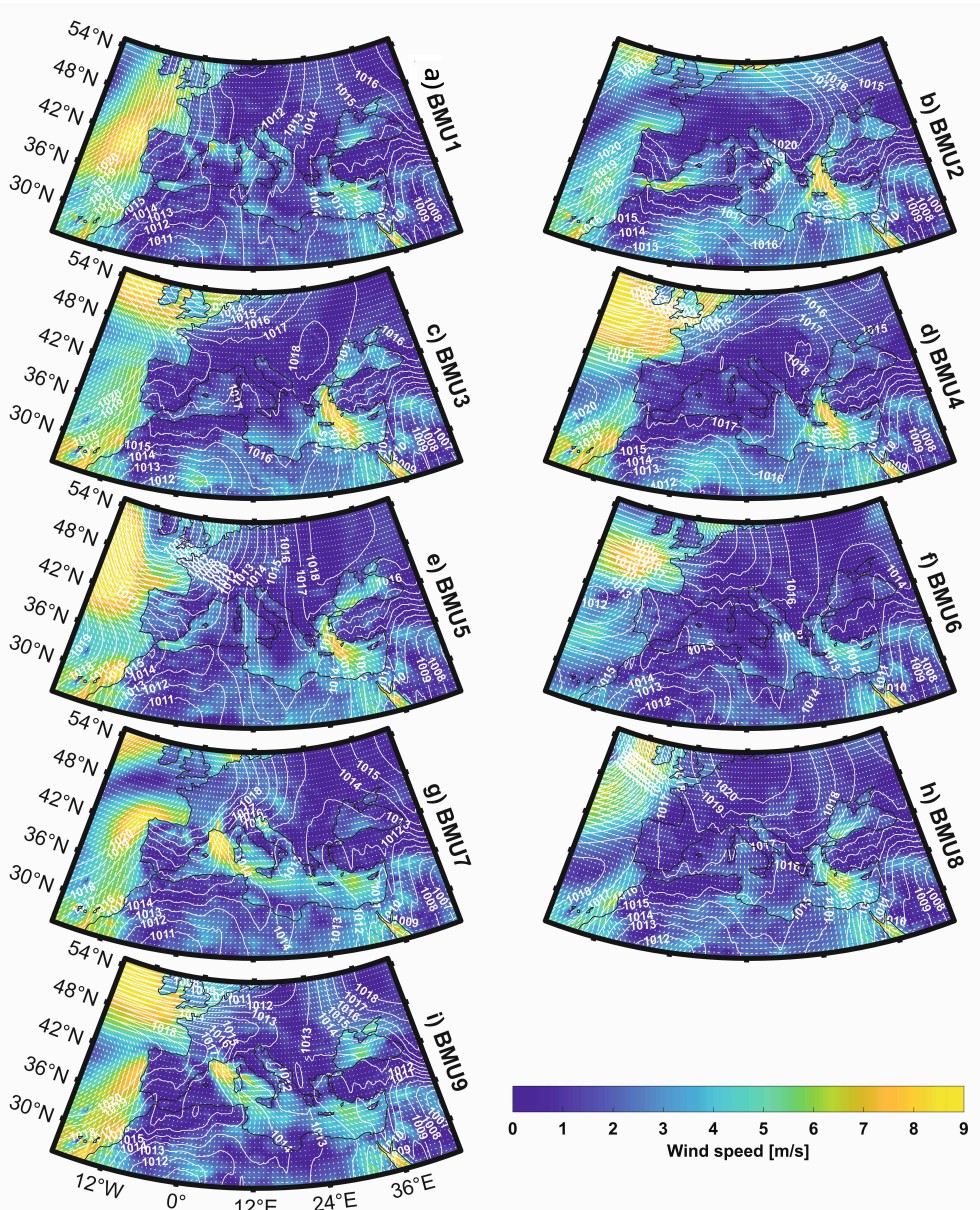

**Figure 5.** Winning neurons (BMUs) calculated with growing neural gas over hourly wind vector data. Isobars (white lines) and sea level mean air pressure fields are smoothed using a Gaussian-weighted moving average over each window of 20 points. The wind speed color bar is the same for all subfigures.

During the period 1980–2020, the appearance of BMUs was not uniform but varied between monthly and decadal time scales (Figures 6 and A2), with no change in stationarity over a period longer than 5 years. To simplify the analysis, we divided the BMUs into four clusters with different wind patterns: SE wind (BMU 1 and 5) as cluster CL1, cyclonic activity (BMU 7 and 9) as cluster CL2, low velocity (BMU 3 and 4) as cluster CL3, and NE–NW wind (BMU 2, 6, and 8) as cluster CL4. The frequencies of CL1–CL4 clusters were 19.2%, 21.3%, 22.6%, and 36.9%, respectively. A significant trend ($p > 0.1$) was observed in most of the clusters. Cyclonic activity increased over the last 40 years, from 120.4 h in 1980 to 184.7 hours in 2020, but no linear trend was observed in the occurrence of SE wind. The frequency of the NE–NW wind pattern decreased from 284.5 h in 1980 to 247.8 h in 2020, while the low speed pattern decreased from 178.8 h in 1980 to 146.3 h in 2020. The quality of the neural gas model can be verified using the heat exchange component over the Adriatic. BMU3 (Table 2) has the highest value for solar radiation and the lowest latent

and sensible heat flux associated with a windless day. The lowest solar radiation is found in BMU5 (SE wind) when the cyclone approaches the Adriatic. The strongest evaporation (BMU 2, 6 and 8) is caused by bora wind outbreaks.

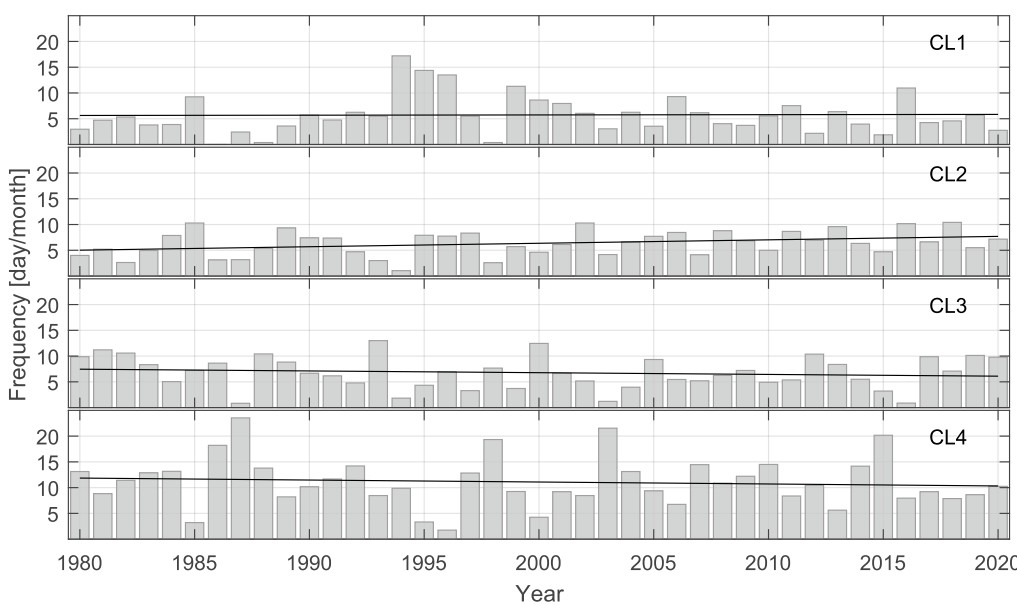

**Figure 6.** The frequency of BMU clusters (grey bars) for the period 1980–2020 for the growing neural gas models grouped for the Adriatic Sea as follows: (CL1) SE wind, (CL2) cyclonic pattern, (CL3) low velocity pattern, and (CL4) NE–NW wind pattern. The linear trend is represented by the black line, with significant ($p < 0.1$; [47,48]) slopes of 0.0052 day/month, 0.067 day/month, −0.034 day/month, and −0.038 day/month for CL1–4, respectively.

**Table 2.** Mean values of heat fluxes (total heat flux ($Q$), incoming solar radiation ($Q_{Sh}$), outgoing longwave radiation ($Q_{Lo}$), latent energy ($Q_{La}$), and sensible energy ($Q_{Se}$)) calculated for modelled best matching units (BMUs) using growing neural gas. Values in bold are statistically different from means according to Student's *t*-test for unpaired samples with $p < 0.01$ (*) or $p < 0.05$ (**). The Fisher–Snedecor F-test is performed to test for equality of variances. If the variances are not equal, Satterthwaite's approximate *t*-test is performed.

|          | BMU1   | BMU2       | BMU3       | BMU4   | BMU5       | BMU6    | BMU7   | BMU8        | BMU9       |
|----------|--------|------------|------------|--------|------------|---------|--------|-------------|------------|
| $Q$      | 438.89 | 461.16     | **495.67 *** | 437.48 | 416.54     | 418.13  | 440.80 | 426.88      | **473.61 *** |
| $Q_{Sh}$ | 627.46 | 634.67     | **661.58 *** | 621.39 | **594.34 **** | 612.60  | 613.06 | 614.41      | **654.68 *** |
| $Q_{Lo}$ | −81.68 | −84.30     | −82.30     | −79.82 | −78.42     | −80.55  | −78.89 | **−77.76 **** | −83.89     |
| $Q_{La}$ | −99.02 | **−82.70 **** | **−79.09 **** | −97.01 | −91.66     | −104.00 | −86.39 | −100.67     | −91.75     |
| $Q_{Sh}$ | −7.87  | −6.50      | **−4.52 **** | −7.09  | −7.72      | −9.93   | −6.99  | −9.10       | −5.44      |

The ocean exchanges energy with the atmosphere through processes of radiative cooling/heating and turbulent energy exchange. In the Adriatic, ocean warming begins in late March and is strongest in the summer months of June, July and August (Figures 7 and 8). The amount of energy received is one of the factors controlling the temperature of the sea surface and the mixed layer. Sensible radiation contributed least, while latent energy varied the most on the seasonal and multi-year scales. The average energy lost by longwave, latent, and sensible energy was 78.85 W/m², 75.63 W/m², and 1.91 W/m², respectively, which was in agreement with [50,51]. A significant linear increase of 14.47 W/m²/decade was seen in the incoming solar radiation, which is probably connected with changes in cloudiness and synoptic patterns over the Adriatic Sea. The changes in wind patterns were responsible for the two climate regimes detected for latent heat. In the period up to 2000,

the sea was losing 211.9 W/m$^2$; meanwhile, in the second regime (2000–2018), the loss was 245.2 W/m$^2$. Due to changes in latent heat and incoming solar radiation, the intensive latent heat loss was compensated for by increased solar radiation. During our research period, the total received energy and latent heat were in the same climate regime, although a significant linear trend was observed in the data. The detailed insight was achieved for all heat exchange components during September in the period of ocean surveys. Compared to the summer period, ocean warming decreased, but the Adriatic Sea still received 450 W/m$^2$ of energy on average. Shifting from mean summer to September daily time scale, the linear trend and climate regimes were not found.

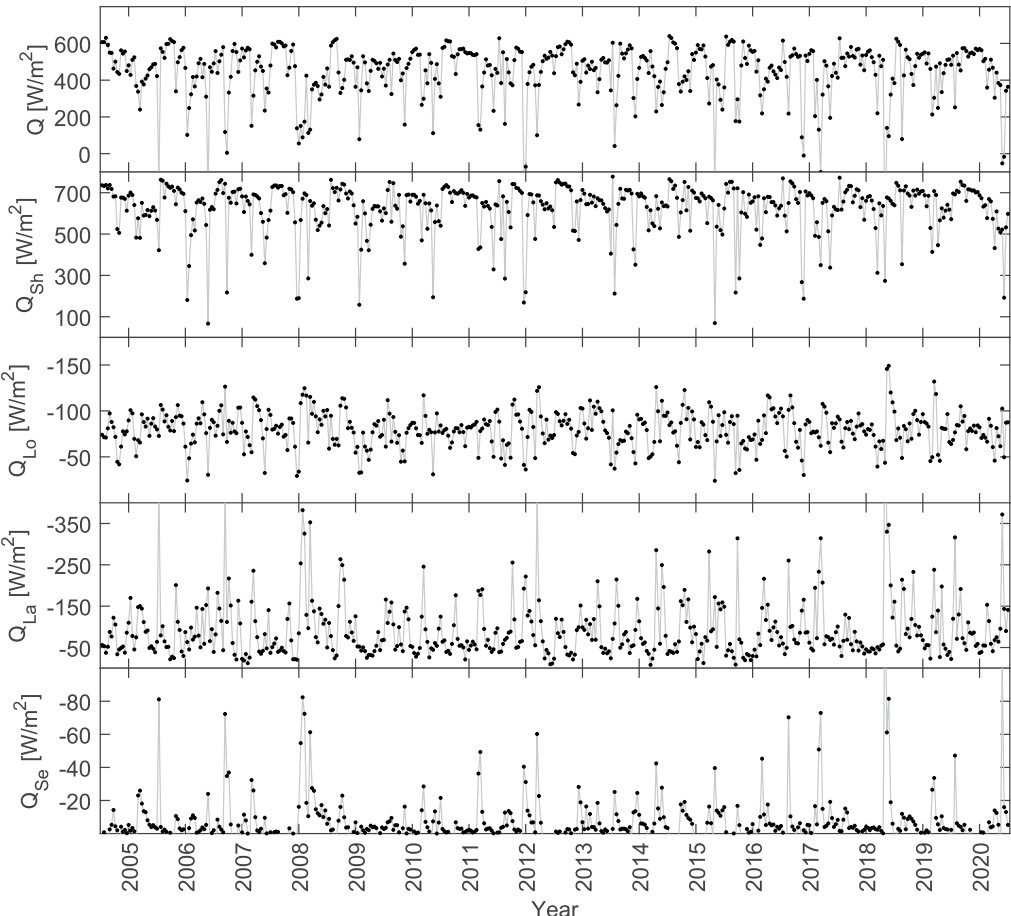

**Figure 7.** Daily fluxes in 12:00 for ERA5 (43.25° N, 14.75° E) wet point total heat flux ($Q$), incoming solar radiation ($Q_{Sh}$), outgoing longwave radiation ($Q_{Lo}$), latent energy ($Q_{La}$), and sensible energy ($Q_{Se}$) for Septembers in the period 2005–2020.

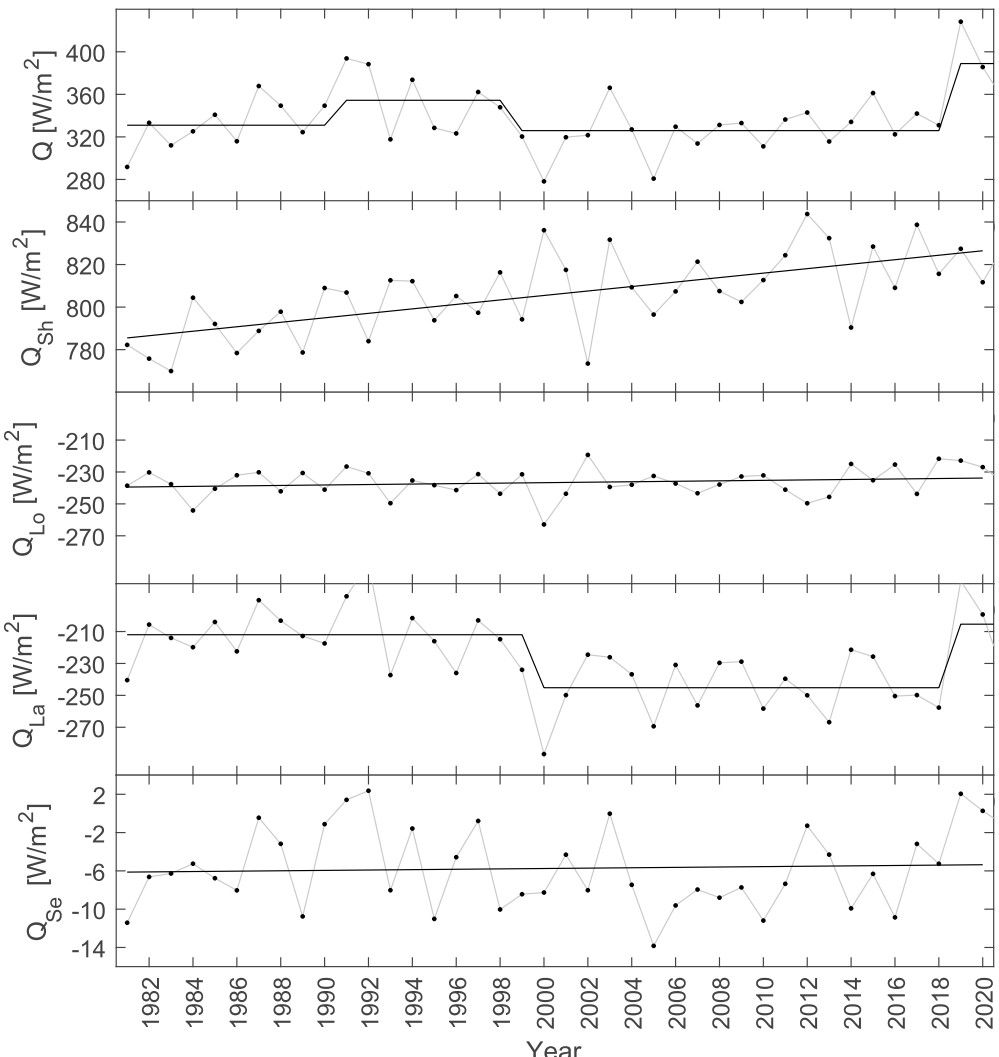

**Figure 8.** Cumulative fluxes in June, July, and August: $Q$ total heat flux, $Q_{Sh}$ incoming solar radiation, $Q_{Lo}$ outgoing longwave radiation, $Q_{La}$ latent energy, and $Q_{Se}$ sensible energy, for the period 1981–2020 for ERA5 (43.25° N, 14.75° E) wet point. The bold line for $Q$ and $Q_{La}$ shows the climate regime ($p < 0.1$, $L = 5$) of the time-series, while for $Q_{Sh}$, $Q_{Lo}$, and $Q_{Se}$ a linear trend can be seen, which is significant ($p < 0.1$) only for $Q_{Sh}$.

## 4. Discussion and Conclusions

A total of 1136 different time-validated and homogenised CTD measurements covering most of the Adriatic Sea for the period 2005–2020 were used to analyse the physical properties of the sea surface layer in September, based on the depth of the mixed layer, depth of the isothermal layer, heat storage, potential energy anomaly, surface barrier layer and wind types, and air–sea energy exchange. The time-series for the depth of the mixed layer showed non-stationarity, with a significant shift in the climate regimes, resulting in the absence of a significant linear trend. In contrast, the mean salinity below the MLD showed a significant linear trend. Three different climate regimes were found for the MLD, HS, and PEA time-series during the studied period, but the one that included the period 2011–2017 was the most interesting (Figures 3 and 4). The reason for this was the increase in HS values during this period and the apparent similarity to the BiOS oscillation [7]. The onset of the regime change in 2011 coincided with the positive phase of BiOS oscillation, whose cyclonic circulation is responsible for the advection of warm and salty water into the Adriatic Sea [8]. The MLD and PEA time-series had a 2-year lag compared to HS, but they also showed a positive trend, although their regime was 2 years shorter. The BiOS

oscillation started in 2018, with a negative phase that abruptly led to an end of the climate regimes for MLD, HS, and PEA, thus proving to be one of the dominant factors affecting stratification in the Adriatic in summer and the associated heat storage.

The reason for the detected anomalies in the observed thermohaline time-series was sought in the regional and local meteorological and oceanographic processes during summer and September (Figures 6–8 and Table 2). As the thermohaline changes persisted over a long period of time, we first examined the regional meteorological processes during the measurement period. Wind patterns over the Mediterranean and the eastern Atlantic showed an increase in cyclonic activity over the Adriatic Sea with decreasing Etesian winds. Cyclonic activity in the Adriatic was associated with cyclonic wind patterns, high precipitation, and a deepening of the MLD. The frequency of calm and Etesian wind patterns decreased during the period 2005–2020, leading to a reduction in incident solar radiation and, consequently, to less warming of the sea surface.

We found significant changes in the interannual variability of salinity above and below the mixed layer depth, in addition to changes in vertical structure (Figure 4). At MLD, a linear increase in salinity of 0.2 PSU/year was observed, related to a decrease in summer precipitation, high evaporation, and the advection of saline water from the Ionian Sea trough during the cyclonic BiOS phase. We found and demonstrated an inverse vertical salinity profile that resulted in a positive difference in the surface boundary layer in 2015, 2017 [5,7], and 2020 (Figure 4c). Together, all of these changes indicate that salinity is the main factor affecting the depth of the boundary layer in certain years.

Finally, this article highlights the utility and importance of acoustic surveys (e.g., EU-MEDIAS) not only for fisheries research, but also as a platform for broader studies of changes in the marine pelagic ecosystem that could serve as a basis for future research into the relationships between marine organisms and their changing environment due to climate change.

**Author Contributions:** Conceptualisation, F.M. and H.K.; methodology, F.M., H.K. and L.Ć.; software, L.Ć.; formal analysis, F.M., T.D., L.Ć., D.S. and L.R.; investigation, F.M. and T.D.; resources, V.T.; data curation, F.M., L.Ć., T.D, T.J. and D.U.; writing—original draft preparation, F.M., T.D, L.Ć. and D.U.; writing—review and editing, F.M., T.D., H.K., L.Ć. and D.U.; visualisation, F.M. and T.D.; supervision, H.K. and F.M.; project administration, H.K.; funding acquisition, H.K. and V.T. All authors have read and agreed to the published version of the manuscript.

**Funding:** This research was partially supported by Croatian Science Foundation (UIP-2019-04-1737) and (IP-2018-01-9849) and the project CAAT "Coastal Auto-purification Assessment Technology" funded by European Union from European Structural and Investment Funds 2014–2020, Contract Number KK.01.1.1.04.0064, and the annual funds for institutional financing of scientific activity from Ministry of Science and Education of Republic of Croatia.

**Institutional Review Board Statement:** Not applicable.

**Informed Consent Statement:** Not applicable.

**Acknowledgments:** The data used in this study were collected during acoustic surveys conducted within framework of national projects PELMON (2005–2012) and EU Mediterranean Acoustic Surveys (EU-MEDIAS: 2013–2020), carried out by Institute of Oceanography and Fisheries and supported by Croatia's Ministry of Agriculture.

**Conflicts of Interest:** The authors declare no conflict of interest.

## Appendix A

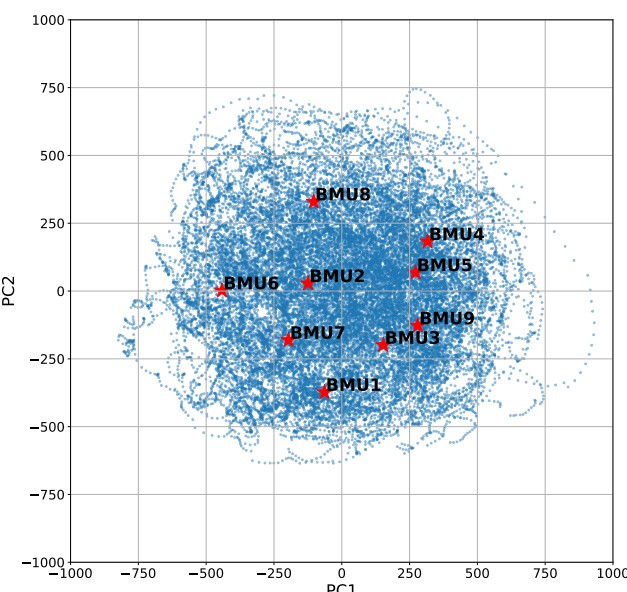

**Figure A1.** 1979–2019 era data plotted onto PC1–PC2 grid. Blue dots represent the raw data, while red stars represent BMUs generated by GNG for the same data set. Each BMU is labeled accordingly.

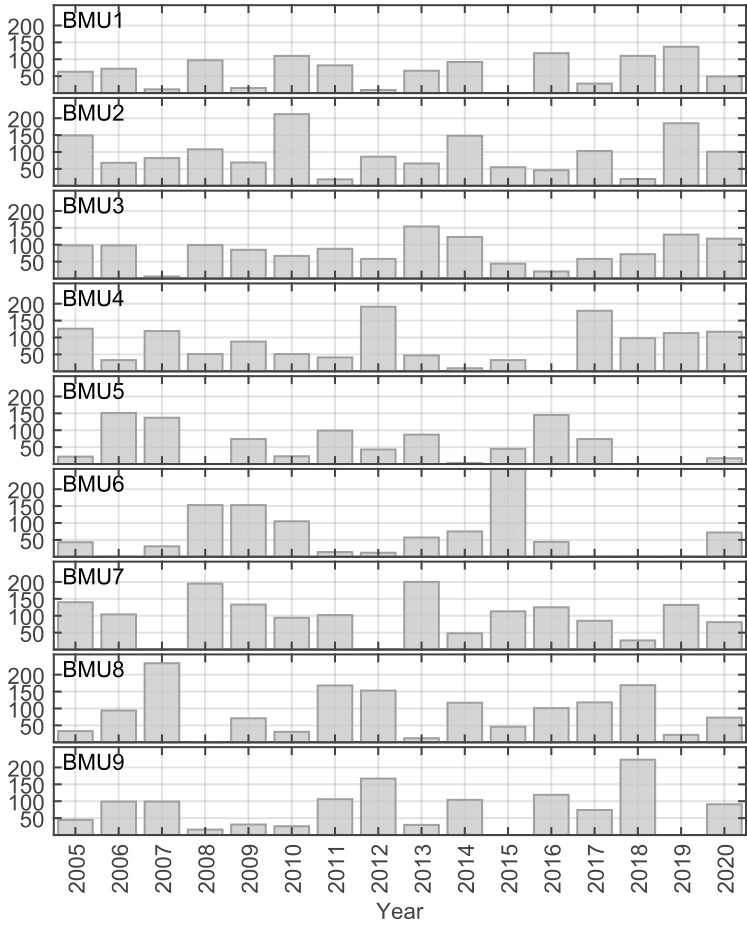

**Figure A2.** Appearance of BMU solutions for the period 2005–2020 for the growing neural gas models.

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
