# Peer review of "Observation of Abrupt Changes in the Sea Surface Layer of the Adriatic Sea"

_jmse, doi:10.3390/jmse10070848_

Round 1
Reviewer 1 Report
Review of “Observation of abrupt changes in the sea surface layer f the Adriatic Sea” by Frano Matić et al.
The authors conducted the CTD observations in the Adriatic Sea for a period between 2005 and 2020 repeatedly and interpreted that changes of the mixed layer’s profile were sometimes caused by the abrupt changes in salinity due to the combination of bimodal oscillation (BiOS) and local factors (e.g., wind fields). The authors examined the CTD observation data, and the results suggested that the mixed layer depth (MLD), heat storage (HS) and potential energy anomaly (PEA) can correlate to the climate regime. Overall the article is well organized and may be published, once the following concerns will be addressed.
General comments:
First of all, I could not understand the motivation of the present study while I was reviewing the article. Can the authors clarify the objective of the present study? The background associated with the regional meteorological changes characterizing sea level or underwater profile of salinity and temperature is mentioned referring to the literatures sufficiently. However, it is not clear what the objective of the present study is. I recommend that a short wording should be included at the last paragraph in the Introduction section.
And then, can you add the place names mentioned in the article to the map (Figure 1)? I could not follow the explanation because the place names I had not heard before were appeared suddenly. I think this may stress other readers, so I would suggest that Figure 1 should be revised if necessary.
Line comments:
Line 82: What is “NAO”? Could you please provide the original terminology of this abbreviation?
Lines 94 – 96: You used the same filter to the all acquired dataset in the data processing. Can you provide more details for data processing e.g., the time-window of the median filter to remove a spike noise? I could not understand what “various quality control procedures” mean.
Lines 127 – 135: Although this part describes the data processing, it is difficult to follow. Could you please show either the flow chart or the block chart corresponding to this description?
Lines 155 & 181: What do “SSS” and “ECMWF” mean? Could you provide the original terminologies of these abbreviations?
Lines 189 – 192: Does Figure 2 really suggest what you described? I cannot understand the association between the temperature – salinity diagram and the description. Where are the Jabuka Pit and the Strait of Otrano in Figure 1?
Line 194: Can you indicate the Velebit Mountains on the inset map of Figure 1?
Lines 204 – 210: Please refer to Figure 3 somewhere, if this paragraph is relevant to Figure 3.
Lines 249 – 250: Where are the Tyrrhenian Sea and the Gulf of Lyon in Figure 1?
Line 398: Continental shelf research > Continental Shelf Research?
Lines 498 & 505: Ocean science > Ocean Science?
Figure 1: I could not imagine the BiOS mechanism mentioned lines 44 – 45, for example, unless all names of the sea are present. Most potential readers may be not familiar with the study area. Could you please indicate the Ionian Sea and the Gulf of Genoa at least in the figure?
Figure 2: Some of plots are forming with small clusters separately from the most plots. Can you add the specific years to the diagram with arrows, if they are associated with the episodic MLD changes or the climate regime?
Figures 3 & 4: If you name each box as (a), (b) and (c), you should indicate (a), (b) and (c) in each box.
Figure 5: If a map with the same scale to the BMUs is available here, it would be helpful for readers.
Figure 7: Do you refer to this figure in the article?
Table 1: I think one “Year” should be removed at the first line of the left column.
Table 2: Do you refer to this table in the article?
Author Response
Response to Reviewer 1 Comments
The authors conducted the CTD observations in the Adriatic Sea for a period between 2005 and 2020 repeatedly and interpreted that changes of the mixed layer’s profile were sometimes caused by the abrupt changes in salinity due to the combination of bimodal oscillation (BiOS) and local factors (e.g., wind fields). The authors examined the CTD observation data, and the results suggested that the mixed layer depth (MLD), heat storage (HS) and potential energy anomaly (PEA) can correlate to the climate regime. Overall the article is well organized and may be published, once the following concerns will be addressed.
Thank you very much for your comment and suggestion.
General comments:
First of all, I could not understand the motivation of the present study while I was reviewing the article. Can the authors clarify the objective of the present study? The background associated with the regional meteorological changes characterizing sea level or underwater profile of salinity and temperature is mentioned referring to the literatures sufficiently. However, it is not clear what the objective of the present study is. I recommend that a short wording should be included at the last paragraph in the Introduction section.
We add the paragraph of the text in the end of Introduction section.
And then, can you add the place names mentioned in the article to the map (Figure 1)? I could not follow the explanation because the place names I had not heard before were appeared suddenly. I think this may stress other readers, so I would suggest that Figure 1 should be revised if necessary.
We have changed the figure 1 adding the toponym (Jabuka Pit, Strait of Otrano, Velebit Mountains, Tyrrhenian Sea and the Gulf of Lyon, Ionian Sea and the Gulf of Genoa).
Line comments:
Line 82: What is “NAO”? Could you please provide the original terminology of this abbreviation?
We defined the abbreviation NAO - North Atlantic Oscillation.
Lines 94 – 96: You used the same filter to the all acquired dataset in the data processing. Can you provide more details for data processing e.g., the time-window of the median filter to remove a spike noise? I could not understand what “various quality control procedures” mean.
We used the time-window of the median filter with size of 10 seconds (80 bins) on depth, temperature and salinity. We added the paragraph in the manuscript.
Lines 127 – 135: Although this part describes the data processing, it is difficult to follow. Could you please show either the flow chart or the block chart corresponding to this description?
We added the paragraph in the manuscript describing the data processing.
Lines 155 & 181: What do “SSS” and “ECMWF” mean? Could you provide the original terminologies of these abbreviations?
We corrected ECMWF is the European Centre for Medium-Range Weather Forecasts. We removed SSS.
Lines 189 – 192: Does Figure 2 really suggest what you described? I cannot understand the association between the temperature – salinity diagram and the description. Where are the Jabuka Pit and the Strait of Otrano in Figure 1?
We have corrected Figure 1. Figure 2 refers to temperature and salinity data collected from 2005 to 2020, and thus is representative of the finding of termohaline properties of the Adriatic Sea. We have corrected Figure 2 and marked the characteristic water masses.
Line 194: Can you indicate the Velebit Mountains on the inset map of Figure 1?
We corrected the Figure 1.
Lines 204 – 210: Please refer to Figure 3 somewhere, if this paragraph is relevant to Figure 3.
We refer the figure 3.
Lines 249 – 250: Where are the Tyrrhenian Sea and the Gulf of Lyon in Figure 1?
We corrected the Figure 1.
Line 398: Continental shelf research > Continental Shelf Research?
We corrected the reference.
Lines 498 & 505: Ocean science > Ocean Science?
We corrected the reference.
Figure 1: I could not imagine the BiOS mechanism mentioned lines 44 – 45, for example, unless all names of the sea are present. Most potential readers may be not familiar with the study area. Could you please indicate the Ionian Sea and the Gulf of Genoa at least in the figure?
We corrected the Figure 1.
Figure 2: Some of plots are forming with small clusters separately from the most plots. Can you add the specific years to the diagram with arrows, if they are associated with the episodic MLD changes or the climate regime?
We corrected the Figure 2 labelling the characteristic water masses.
Figures 3 & 4: If you name each box as (a), (b) and (c), you should indicate (a), (b) and (c) in each box.
We corrected the Figure 3 & 4.
Figure 5: If a map with the same scale to the BMUs is available here, it would be helpful for readers.
We corrected the figure caption adding the text: The wind speed colour bar is the same for all subfigures.
Figure 7: Do you refer to this figure in the article?
We corrected.
Table 1: I think one “Year” should be removed at the first line of the left column.
We corrected the Table 1.
Table 2: Do you refer to this table in the article?
We cited the Table 2.
Reviewer 2 Report
General comments and questions
Did the authors consider the possibility of conducting this study with the usage of other products of in situ observations like those from Copernicus Marine Service (CMEMS), EN4, World Ocean Database? Beyond CTD, these datasets contain in situ T/S measurements from different platforms (e.g. Argo) that cover the Adriatic Sea, which could bring more robust results. Why consider only T/S observations from CTD probes from acoustic surveys?
Due to the scarcity of observations, did the authors think about using model or reanalyses results?
Is 0.25o x 0.25o the highest spatial resolution available for ERA5? Why the decision of using ERA5 with this resolution?
The conclusions are unclear and require a major review.
Specific comments
2. Materials and Methods
2.2.2 Sea data analysis method
(line 170) Would be cp the specific heat capacity?
(line 185) The abbreviation “RS” is initially mentioned at this point, which is RS?
3. Results
Table 2 is not cited along the text.
Both Figures 6 and 7 are not cited in the text.
Figure 2. In the T/S diagram, would it be possible to separate the profiles by color according to the season (winter, spring, summer, fall)? It could facilitate the understanding of the first paragraph (lines 188-203) in the Results section.
(lines 204-210) This paragraph seems to describe the plots in Figure 3, but no figure is mentioned.
(lines 222-224) My suggestion is to rewrite the sentence as follows: “In the Adriatic and Mediterranean Sea, the influence of salinity on SBL is less important than that of temperature, resulting in values around zero most of the time”.
(line 225) “The main result was slightly positive SBL values”. I do not see it as a main result, why? Please explain it.
(lines 233-237) Where is the plot/figure showing the thermocline variations? Please cite it.
(lines 246-248) Improve the representation of wind vectors in Figure 5. Perhaps it may be convenient to mask the wind in the Atlantic.
(lines 256-263) Is there a reason to move the discussion from Figure 5 directly to Figure 9?
(lines 264-275) Since the wind vectors are not well-represented in Figure 5, it has been difficult to understand how the BMUs were divided into four clusters according to the wind patterns.
(page 12) Verify the title of the y-axes in Figure 8, sensible energy (Qse) is missing.
3. Discussion and Conclusions
(lines 304-305) I consider important to mention Figure 3 here.
It is important to mention the Figures or it is difficult to follow the discussion.
Author Response
Response to Reviewer 2 Comments
Thank you very much for your comment and suggestion.
Did the authors consider the possibility of conducting this study with the usage of other products of in situ observations like those from Copernicus Marine Service (CMEMS), EN4, World Ocean Database? Beyond CTD, these datasets contain in situ T/S measurements from different platforms (e.g. Argo) that cover the Adriatic Sea, which could bring more robust results. Why consider only T/S observations from CTD probes from acoustic surveys? Due to the scarcity of observations, did the authors think about using model or reanalyses results?
At the beginning of the preparation of the manuscript, we searched for all available free temperature and salinity data, both measured (with different instruments) and modelled. We found that the temporally and spatially sparse Argo floats and CTD measurements were not suitable for our analysis. Therefore, we focus on the largest collection (in a spatial sense) of data collected in September. Another reason for analysing the data collected during echo monitoring is to investigate ways to remotely estimate the vertical sound velocity profiles required for the survey protocol. Of course, we have analysed the MLD from the Copernicus reanalysis database (https://doi.org/10.25423/CMCC/MEDSEA_ANALYSISFORECAST_PHY_006_013_EAS6), but it has been shown that the data only roughly match the measured data.
Is 0.25o x 0.25o the highest spatial resolution available for ERA5? Why the decision of using ERA5 with this resolution?
In using ERA5 data, we focus on the representativeness of the flux data over the sea. We found that spatial variations are less important than the correct selection of the ERA5 wet point (doi:10.3390/s21103507). As noted in the manuscript, we used the ERA5 wet point 43.25 N, 14.75 E.
The conclusions are unclear and require a major review.
As suggested in your review, we mention the figures and tables supporting the results in the Conclusions section. In response to your comments, we try to clear up the misunderstandings related to the conclusion section.
Specific comments
(line 170) Would be cp the specific heat capacity?
We corrected the cp to cp is the specific heat capacity of the seawater.
(line 185) The abbreviation “RS” is initially mentioned at this point, which is RS?
We defined the abbreviation RS as regime shifts.
Table 2 is not cited along the text.
We cited the Table 2.
Both Figures 6 and 7 are not cited in the text.
We cited the Figure 6 & 7.
Figure 2. In the T/S diagram, would it be possible to separate the profiles by color according to the season (winter, spring, summer, fall)? It could facilitate the understanding of the first paragraph (lines 188-203) in the Results section.
We corrected the Figure 2 labelling the characteristic water masses.
(lines 204-210) This paragraph seems to describe the plots in Figure 3, but no figure is mentioned.
We refer the figure 3.
(lines 222-224) My suggestion is to rewrite the sentence as follows: “In the Adriatic and Mediterranean Sea, the influence of salinity on SBL is less important than that of temperature, resulting in values around zero most of the time”.
We accepted your suggestion.
(line 225) “The main result was slightly positive SBL values”. I do not see it as a main result, why? Please explain it.
At the end of abstract (L18-19) and in the conclusion section there is the paragraph L346-L353, which shows the effect of positive SBL.
(lines 233-237) Where is the plot/figure showing the thermocline variations? Please cite it.
We rewrite the sentence. We assumed thermocline variations as MLD variations.
(lines 246-248) Improve the representation of wind vectors in Figure 5. Perhaps it may be convenient to mask the wind in the Atlantic.
The purpose of the neural gas analysis (Figure 5 and Table 2) is to study the hemispheric meteorological influence on the local (Adriatic) wind pattern and heat exchange. If only the Adriatic wind pattern were presented, the analysis would go in a direction that is not interesting for this manuscript.
(lines 256-263) Is there a reason to move the discussion from Figure 5 directly to Figure 9?
Figure 9 is a supporting figure, so we have included it in the appendix.
(lines 264-275) Since the wind vectors are not well-represented in Figure 5, it has been difficult to understand how the BMUs were divided into four clusters according to the wind patterns.
We are not sure if we understood your comment correctly, but we never mentioned that the wind vectors are not well represented, on the contrary, in Figure 9 and Table 2 we showed that the neural gas successfully modelled the wind pattern over the studied area and can be associated with the heat exchange component over the Adriatic Sea. The clusters were found by averaging the wind vector over the Adriatic Sea.
(page 12) Verify the title of the y-axes in Figure 8, sensible energy (Qse) is missing.
We corrected Figure 7 and 8.
(lines 304-305) I consider important to mention Figure 3 here. It is important to mention the Figures or it is difficult to follow the discussion.
We mention the Figure 3 and all relevant figures and tables.
Round 2
Reviewer 2 Report
General comments:
I suggest improving the topic "Discussion and Conclusions". In my opinion, the discussion is still poor. Consider comparing the results with previous studies in the Adriatic Sea. If the study is very pioneering and there is not relevant literature to be discussed, it is important to mention it.
I give more information on my previous comment as follows: "Since the wind vectors are not well-represented in Figure 5, it has been difficult to understand how the BMUs were divided into four clusters according to the wind patterns". I mentioned about the superimposed vectors in Figure 5, they are not clear, which makes it difficult for readers to understand the direction and pattern of the wind. Try to improve it.
Author Response
Dear Reviewer, Thank you for reviewing the manuscript and for your comments.
I suggest improving the topic "Discussion and Conclusions". In my opinion, the discussion is still poor. Consider comparing the results with previous studies in the Adriatic Sea. If the study is very pioneering and there is not relevant literature to be discussed, it is important to mention it.
The survey data used in the manuscript are new and cover almost the entire Adriatic Sea over a long period in September. We have cited the relevant publications in which the MLD is treated as a case study.
I give more information on my previous comment as follows: "Since the wind vectors are not well-represented in Figure 5, it has been difficult to understand how the BMUs were divided into four clusters according to the wind patterns". I mentioned about the superimposed vectors in Figure 5, they are not clear, which makes it difficult for readers to understand the direction and pattern of the wind. Try to improve it.
We have analysed the quality of Figure 5 in the submitted manuscript and found that the manuscript has a lower quality figure. We hope that the published manuscript will contain Figure 5 with the highest resolution.